# ZnO Nanoparticles-Modified Dressings to Inhibit Wound Pathogens

**DOI:** 10.3390/ma14113084

**Published:** 2021-06-04

**Authors:** Sajjad Mohsin I. Rayyif, Hamzah Basil Mohammed, Carmen Curuțiu, Alexandra Cătălina Bîrcă, Alexandru Mihai Grumezescu, Bogdan Ștefan Vasile, Lia Mara Dițu, Veronica Lazăr, Mariana Carmen Chifiriuc, Grigore Mihăescu, Alina Maria Holban

**Affiliations:** 1Microbiology & Immunology Department, Faculty of Biology, University of Bucharest, 77206 Bucharest, Romania; sajjadmohsin13@gmail.com (S.M.I.R.); hamza_basil@yahoo.com (H.B.M.); carmen.curutiu@bio.unibuc.ro (C.C.); lia-mara.ditu@bio.unibuc.ro (L.M.D.); veronica.lazar@bio.unibuc.ro (V.L.); carmen.chifiriuc@bio.unibuc.ro (M.C.C.); grigore.mihaescu@bio.unibuc.ro (G.M.); 2Research Institute of the University of Bucharest—ICUB, University of Bucharest, 050657 Bucharest, Romania; grumezescu@yahoo.com; 3Department of Science and Engineering of Oxide Materials and Nanomaterials, Faculty of Applied Chemistry and Materials Science, Politehnica University of Bucharest, 011061 Bucharest, Romania; ada_birca@yahoo.com (A.C.B.); vasile_bogdan_stefan@yahoo.com (B.Ș.V.); 4Academy of Romanian Scientist, Ilfov Str. No. 3, 50044 Bucharest, Romania

**Keywords:** chronic wounds, biofilm control, opportunistic pathogens, ZnO nano-coatings, antimicrobial nanoparticles

## Abstract

Zinc oxide (ZnO) nanoparticles (NPs) have been investigated for various skin therapies in recent years. These NPs can improve the healing and modulate inflammation in the wounds, but the mechanisms involved in such changes are yet to be known. In this study, we have designed a facile ZnO nano-coated dressing with improved antimicrobial efficiency against typical wound pathogens involved in biofilm and chronic infections. ZnO NPs were obtained by hydrothermal method and characterized by X-ray diffraction, scanning electron microscopy, transmission electron microscopy, and Fourier-transform infrared spectroscopy. Antibacterial and antibiofilm effects were evaluated against laboratory and clinical isolates of significant Gram-negative (*Pseudomonas aeruginosa* and *Escherichia coli*) and Gram-positive (*Staphylococcus aureus* and *Enterococcus faecalis*) opportunistic pathogens, by quantitative methods. Our results have shown that the developed dressings have a high antibacterial efficiency after 6–24 h of contact when containing 0.6 and 0.9% ZnO NPs and this effect is similar against reference and clinical isolates. Moreover, biofilm development is significantly impaired for up to three days of contact, depending on the NPs load and microbial species. These results show that ZnO-coated dressings prevent biofilm development of main wound pathogens and represent efficient candidates for developing bioactive dressings to fight chronic wounds.

## 1. Introduction

Wound healing is often complicated by infection caused by mono- or polymicrobial aerobic and anaerobic microorganisms resistant to biocides and having the ability to develop thick biofilms [1]. These specialized multicellular microbial aggregates show increased phenotypic resistance (tolerance) to biocides and host defense mechanisms and drastically reduce the effectiveness of antimicrobial treatments [2]. Different types of biofilms are formed within the wound beds, especially within chronic wounds, representing a physical barrier to wound healing, with the inflammatory phase being extended [3,4,5]. Mature biofilms gradually develop into chronic wounds within 10–12 h and persist while the wound remains open [6]. *Pseudomonas aeruginosa*, *Staphylococcus aureus*, and some enterobacteria species have proved to form difficult to eradicate biofilms in the wounds and are considered the most challenging etiologies in the management of chronic wounds [5,7]. Most of these isolates show modified virulence and are resistant to common antibiotics. The most prevalent resistant etiologies in chronic skin infections are MRSA (methicillin-resistant *S. aureus*), ESBL (Extended Spectrum Beta-Lactamase)-producing enterobacteria, and multiple antibiotic resistant (MDR) *P. aeruginosa* [7,8]. With an increasing incidence of chronic wounds and, implicitly, chronic biofilm infections, there is a need for alternative therapeutic agents and dressings.

For decades, the main aim of wound dressing developers has been to protect and cover the wound from microbial infections [9,10]. The absorbent first-line dressings found on the market can be made of textile fibers, alginate, foam, hydrocolloid, and polysaccharide bead dressings [9,10,11]. Even though some protective actions are shown by the dressings, they still face multiple challenges in terms of managing the exudates, controlling bacterial count, and promoting faster healing [12,13]. Nanotechnology shows promising openings in the design of efficient antimicrobial wound dressings, either by the intrinsic antimicrobial properties of nanoparticles or their function as drug carriers. Metal and metal oxide nanoparticles (NPs) such as gold, silver [14,15], zinc oxide [16], and magnetite can be active at low concentrations toward a large variety of infectious agents [17]; thus, they are unlikely to elicit the emergence of resistance [1,18]. NPs modulate the microbial colonization and biofilm formation in the wounds, as recent studies show [1,11,19]. Metal nanoparticles (e.g., silver, gold, zinc oxide, magnetite) are now being used more in skin applications, including the design of nanostructured dressings and coatings. Irrespective of their metallic nature, such NPs are mostly used owing to their unique properties of being antibacterial and having the ability to penetrate into deep skin [20,21]. NPs may be tailored to integrate and deliver various molecules with therapeutic potential [22,23]. They may be efficient against both susceptible and resistant pathogens and tolerant biofilms [24,25].

Zinc oxide (ZnO) nanoparticles have enhanced antimicrobial properties, being efficient against clinically relevant microbial pathogens, such as *S. aureus* [26], *P. aeruginosa* [27] and *Escherichia coli* [28]. Such NPs were recently investigated for the treatment of wound infections in vitro and in vivo [26]. Although their antimicrobial mechanism is not fully understood, studies show that their photocatalytic abilities and interference with the production of reactive oxygen species, which elevates membrane lipid peroxidation that causes membrane leakage of reducing sugars, DNA, proteins, and reduces viability in Gram-negative bacteria [28], are the main factors. Moreover, ZnO NPs could interfere with microbial virulence, altering the expression of key genes involved in cell stress response, motility, pathogenesis, and toxin production [27]. NPs containing ZnO were also recently investigated for their anti-biofilm activity [28,29] and wound applications in vitro [30]. Recent studies identify the release of Zn^2+^ ions from ZnO NPs as a major mechanism for oligodynamic activities against both eukaryotic and prokaryotic cells. The uncontrollable release of Zn^2+^ ions in certain conditions is considered an important hindrance to obtain appropriate industrial formulation [31]. This leads to the need for a high dose of NPs with the risk of neutralization of the therapeutic effects of formulation and increased side effects and preparation costs. Hydrogel formulations, with smart and sustained release of Zn^2+^ ions by suitable polymers [32], are considered to be efficient alternatives to obtain ZnO NPs bioactive formulations [31].

The purpose of this study was to obtain a ZnO NPs-coated polyester-nylon wound dressing with improved antimicrobial properties and efficiency against recently isolated wound pathogens with biofilm-forming activity. ZnO NPs interactions were previously studied with a few polymers, such as alginate [33], chitosan [34], and cellulose [35]; however, we have selected polyester-nylon dressings since they are widely available and currently utilized in routine wound management. The main novelty of this study is the design of a facile ZnO NPs-containing wound dressings, which can be widely available and efficient against both planktonic and biofilm-embedded resistant strains of most relevant wound pathogens. We have used different concentrations of Zn(II) (0.3, 0.6, and 0.9%) in order to obtain various types of nanomodified dressings and evaluate their antimicrobial efficiency.

## 2. Materials and Methods

### 2.1. Materials

The materials used to prepare the nano-coatings (zinc nitrate, sodium hydroxide, and D-glucose) were procured from Sigma-Aldrich (Saint Louis, Missouri, USA). Being of analytical grade, these materials were used without further purification. Double-distilled water served as the solvent for the preparation of the required aqueous solutions. A local supplier provided the experimentally modified wound dressings, which consisted of commercial polyester-nylon dressings.

Surface-modified wound dressings were prepared as follows: polyester-nylon dressings were cut into round samples of 6 mm diameter and immersed into 3% NaOH (100 mL) solution. The solution was stirred continuously at 70 °C before being added dropwise into different concentrations (0.3, 0.6, or 0.9%) of Zn(NO_3_)_2_ 6H_2_O prepared in 100 mL distilled water (DW) at 70 °C. The obtained ZnO-based dressings were washed with DDW (double distiled water) and dried at room temperature. The samples were noted as ZnO 0.3, Zn 0.6, or ZnO 0.9 based on the Zn(NO_3_)_2_ 6H_2_O content.

### 2.2. Physico-Chemical Characterization of Nano-Coatings

#### 2.2.1. X-ray Diffraction

An X-ray diffraction analysis of nanomaterial powders was performed with a Panalytical Empyrean diffractometer (Panalytical, Malvern, UK) (step size 0.02, time per step 1 s) at room temperature. For all analyses made, Cu Ka radiations with l = 1.541874 Å were used. Samples were scanned at a Bragg 2theta angle between 10–80.

#### 2.2.2. Scanning Electron Microscopy (SEM)

A FEI Quanta Inspect F scanning electron microscope (Thermo Fisher Scientific, Hillsboro, OR, USA) was used to evaluate the morphology of materials. Prior to analysis, samples were capped with a thin gold layer and the SEM micrographs were recorded using secondary electron beams (energy of 30 keV).

#### 2.2.3. Fourier-Transform Infrared Spectroscopy (FTIR)

FTIR analysis was performed by using a Nicolet 6700 FT-IR spectrometer (Thermo Nicolet, Madison, WI, USA). The OMNIC operating system software was Version 8.2; Thermo Nicolet, Madison, WI, USA. The obtained modified wound dressings were placed in contact with attenuated total reflectance (ATR) on a multibounce plate of ZnSe crystal at controlled ambient temperature (25 °C). We have used the frequency range of 4000–650 cm^−1^ to collect the FT-IR spectra by co-adding 32 scans and at a resolution of 4 cm^−1^ with strong apodization. All spectra were ratioed against a background of an air spectrum.

#### 2.2.4. Transmission Electron Microscopy (TEM)

The TEM images were obtained using a Tecnai™ G2 F30 S-TWIN high-resolution transmission electron microscopy from FEI (FEI Company, Hillsboro, OR, USA) equipped with area electron diffraction and selected area electron diffraction. The microscope was operated in transmission mode at 300 kV with TEM point resolution of 2 Å and line resolution of 1 Å. After preparation, the sample was put onto a holey carbon-coated copper grid and left to dry before it was analyzed through TEM.

### 2.3. Antimicrobial Assessment

#### 2.3.1. Bacterial Strains

The antibacterial evaluation of the nanomodified wound dressings was assessed in vitro against 10 clinically relevant bacterial strains previously isolated from wound infections. Five Gram-positive (*S. aureus*, *E. feacalis*) and five Gram-negative (*E. coli* and *P. aeruginosa*) bacteria strains were utilized in this study (Table 1). The strains are maintained as glycerol stocks at −80 °C in the culture collection of the Microbiology-Immunology Department of the Faculty of Biology, University of Bucharest.

#### 2.3.2. Bacterial Viability

Overnight bacterial cultures of the microbial strains were used to prepare microbial suspensions of 0.5 McFarland (approximately 1.5 × 10^8^ CFU/mL) density in sterile phosphate-buffered saline (PBS). Wound dressing specimens were sterilized by UV irradiation for 20 min before analysis. Then, they were added into sterile 24 multi-well plates containing 1 mL of nutritive broth, and 10 µL of the obtained suspension were seeded in each sample. The plates were incubated for 1 h, 6 h, 12 h, and 24 h at 37 °C in a moist atmosphere. For the viability cells assay, after the incubation time, each wound dressing sample was resuspended in 1 mL PBS into an Eppendorf tube to recover the viable microbial cells. Tubes were vigorously vortexed for 20 s to release the microbial cells into the suspension. Then 30 µL of the obtained suspensions were transferred to 96 well plates containing 270 µL of PBS. Serial dilutions were obtained from each sample and then cultured in triplicate on nutrient agar plates for 24 h at 37 °C [36]. The obtained colonies were then counted and the CFU/mL (colony forming units/mL) was achieved.

#### 2.3.3. Planktonic Growth in Nutritive Broth

The effect of the obtained materials on the growth of microorganisms in liquid medium (planktonic cultures) was tested. A piece of sterile material (coating) was individually deposited in a well of a sterile 24-well plate. Over the deposited materials, 1 mL of liquid medium (nutritive broth) and subsequently 10 μL of 0.5 McFarland density microbial suspensions prepared in PBS were added. The prepared plates were incubated at 37 °C for 24 h. Then 150 μL of the obtained microbial culture (planktonic cells) were transferred to 96 well plates and the turbidity of the microbial cultures (absorbance, Abs 600 nm) was measured spectrophotometrically [37].

#### 2.3.4. Monospecific Biofilm Development

In this assay we have evaluated short-term and long-term antimicrobial efficiency against monospecific biofilms. The antibiofilm efficiency was established by evaluating the bacterial biofilm development in the presence of regular polyester-nylon dressings (positive control) and nano-coated wound dressings. The control and nano-modified wound dressing specimens were placed in sterile 24-well plates in 1 mL nutritive broth, followed by the inoculation of 10 μL of microbial suspension of 0.5 McFarland standard density from each bacterial strain. The as-prepared plates were incubated for 24 h at 37 °C. After incubation, culturing media was removed and the samples were washed with 1 mL sterile PBS to remove the unattached bacteria. The wound dressing samples were then transferred to sterile 24-well plates containing fresh media and incubated for 24, 48, and 72 h, respectively, at 37 °C to allow the growth and biofilm formation of the attached bacteria. After incubation, the wound dressing samples were gently washed with sterile phosphate-buffered saline and further placed in 1.5 mL centrifuge tubes containing 1000 μL of PBS. The obtained specimens were vortexed for 30 s and subsequently subjected to ultrasounds for 10 s to detach the biofilm cells and obtain microbial suspensions of cells which were previously embedded into biofilms. Serial ten-fold dilutions were performed from the obtained suspensions and inoculated on nutrient agar for viable cell counts assay [37,38]. All the experiments were performed in triplicate and repeated on three separate occasions.

## 3. Results

### 3.1. Physico-Chemical Characterization

#### 3.1.1. Microscopy Analyses

In this study, we aimed to obtain a bioactive wound dressing coated with ZnO NPs able to reduce biofilm formation in key opportunistic pathogens (i.e., Gram-positive *S. aureus* and *E. faecalis*, and Gram-negative *P. aeruginosa* and *E. coli*) isolated from wound biofilms. Figure 1 reveals the SEM images obtained at various magnifications of the produced ZnO nano-coatings. In all cases, ZnO nanoparticles were present with a diameter varying between 50 and 80 nm. There is a tendency of agglomeration of nanoparticles and it can be seen that the coatings are non-uniform. It seems that at concentrations of 0.3% and 0.6% the agglomeration is increased and the fibers are less uniformly covered, while by increasing the concentration (at 0.9%) of ZnO NPs, the agglomeration is reduced and the fiber is covered in a more uniform manner. The amount of nanoparticles that interact with the dressings increases with the concentration of Zn(II) (Figure 1).

Figure 2 presents micrographs obtained after TEM evaluation of the ZnO NPs. This analysis highlights an average size of 40 nm and quasi-spherical morphology of ZnO NPs. At high resolution, single-phase hexagonal ZnO was identified. This result is in good agreement with XRD and SAED pattern.

#### 3.1.2. FT-IR

IR spectra of prepared samples and control (uncoated ZnO dressing) are plotted in Figure 3. It can be seen that the functional groups available in the dressing are also available in the coated ZnO dressings in the case of all samples: 0.3, 0.6, and 0.9%. The identified groups were OH at 3324 cm^−1^, C–H at 2915 and 2862 cm^−1^, and C=O at 1712 cm^−1^. Also, absorbances available at 1081 and 1017 are available to C–O bonds. Zn–O bond was identified at 425 cm^−1^. It can be seen that Zn–O absorbance is available only in the samples ZnO 0.6 and ZnO 0.9%, while ZnO 0.3 has no visible Zn–O absorbance. This is due to the low amount of ZnO in the sample under the equipment’s detection limit. Even if IR did not allow to identify the Zn–O in the ZnO 0.3 sample, SEM analysis highlighted the presence of ZnO nanoparticles on the surface of coated ZnO 0.3 dressing.

#### 3.1.3. XRD (X-ray Diffraction)

The prepared ZnO NPs-coated dressings were characterized by XRD. The XRD patterns are depicted in Figure 3. The XRD patterns shows peaks attributed to the ZnO at 2θ values of 31.79° (1 0 0), 34.44° 2 (0 0 2), 36.27° (1 0 1), 47.56° (1 0 2), 56.61° (1 1 0), 62.87° (1 0 3), and 67.96° (1 1 2), and they were indexed to different planes of crystalline ZnO as previously described by [39]. The ZnO NPs (available on the surface of dressings) obtained in the present work were single-phase hexagonal ZnO (Figure 4), similar with the study of [40]. It can be seen that ZnO 0.3 sample has no peaks in the XRD pattern. This is due to the low amount of ZnO in the sample, which is under the detection limit of the equipment. These results are well corelated with FT-IR analysis. The Zn–O bond in the sample ZnO 0.3 cannot be identified for the same reason.

### 3.2. Antimicrobial Evaluation

#### 3.2.1. Viability in PBS

In order to analyze the antimicrobial effect of the tested wound dressings, we have performed several growth and viability tests in PBS. Viability in PBS was assessed at 4 time points, respectively 1 h, 6 h, 12 h, and 24 h, in order to establish the bacteria-killing intrinsic potential of the nano-coated dressings.

We have observed that microbial viability is impaired in a time- and ZnO NPs-dependent manner. Significant viability loss in the presence of ZnO-coated dressings was observed after at least 12 h of incubation in the case of Gram-positive analyzed strains (i.e., *S. aureus* and *E. faecalis*) (Figure 5), while in Gram-negative isolates viability was significantly decreased for all three *P. aeruginosa* tested strains and one *E. coli* strain after 6 h of incubation (Figure 6). Another difference between Gram-positive and Gram-negative bacterial strains viability in the presence of coated dressings is the concentration of ZnO NPs on the coatings. Therefore, in care of Gram-positive strains, the coatings containing concentrations of 0.6% and 0.9% ZnO NPs induced significant viability loss after 12 h of incubation, while after 24 h, viability loss was achieved for all tested ZnO NPs concentrations (0.3, 0.6, and 0.9%) (Figure 5). On the other hand, in Gram-negative selected strains, especially *P. aeruginosa*, significant viability reduction is observed after 6 h of incubation in the presence of all tested ZnO NPs-coated dressings, regardless their NPs concentration (Figure 6).

Differences in Gram-positive and Gram-negative viability loss in the presence of ZnO nano-coated dressings could be related to variations in the cell wall of the two bacteria groups. It is recognized that the main bacteria-killing mechanism exhibited by ZnO NPs is correlated to permeability changes and cellular membrane integrity loss [41,42,43]. Because of their cell wall structure (the presence of an additional outer membrane and a thin peptidoglycan layer) [44], Gram-negative bacteria may be more susceptible to cellular wall piercing by ZnO NPs.

#### 3.2.2. Planktonic Growth

Planktonic (free-floating) growth of microorganisms is important in the evaluation and management of wound infections. It has been recently suggested that high-density bacterial growth is critical for infection and chronic wound progression. This aspect seems to be more important than bacterial virulence and even biofilm formation for the fitness of infecting bacteria in chronic wounds [45]. Our planktonic growth results demonstrated that the tested nano-coated dressings present different antimicrobial effects depending on the ZnO NPs concentration. Thus, after 24 h incubation in nutritive broth in the presence of nano-coated dressings, all tested strains show significantly impaired growth when concentrations of 0.6% and 0.9% ZnO NPs are used (Figure 7). However, a slight difference between Gram-positive and Gram-negative bacteria strains planktonic growth was also observed, the results being consistent with viability data.

However, even less intense, the inhibition of planktonic growth in *S. aureus* and *E. faecalis* strains is still significant for all ZnO assessed concentrations (growth in the presence of ZnO (Abs 600 nm is ~0.12) is about half the growth of control dressing (Abs 600 nm is ~0.26)). Also, the great inhibition of planktonic growth in *E. coli* and *P. aeruginosa*, which are known for their multiple resistance rates [46], represents a very relevant finding of our study, suggesting these coatings could be very efficient in inhibiting resistant pathogens.

#### 3.2.3. Biofilm Modulation

Biofilm growth is seen in many of the chronic wound infections. Bacteria embedded into thick biofilms become resistant to virtually any known antibiotic and traditional biocide which can be applied locally for chronic wound management [47]. For this reason, our approach was to cover both planktonic and biofilm bacteria growth in order to obtain a clear idea regarding the coverage of the antimicrobial activity of the obtained coatings. The data have shown that biofilms formation is also inhibited in the presence of nano-coated wound dressings. We have found that 24 h biofilms are most significantly inhibited in a NPs dose-dependent manner, the most significant inhibition being observed for 0.9% ZnO, followed by 0.6% ZnO content in all microbial strains (Figure 8). Significant biofilm inhibition is maintained for 48 h for all tested microbial species at 0.6% and 0.9% ZnO NPs-coated dressings, while only *P. aeruginosa* biofilms seem to be significantly reduced after 72 h incubation in the presence of tested nano-coatings (Figure 8). This result suggests that the bioactivity of the ZnO nano-coated dressings is maintained for at least 48 h for most important wound biofilm pathogens (both laboratory and clinical resistant isolates) and it may be prolonged for particular Gram-negative bacteria, such as the opportunistic pathogen *P. aeruginosa*.

The fact that the results showed significant biofilm inhibition in relevant pathogens responsible for biofilm and chronic wounds supports the idea that ZnO nano-coated dressings could be utilized as efficient tools in the management of difficult wounds. Also, their efficiency was similar in laboratory and clinical isolates of the analyzed species and this demonstrates that such nano-coatings could cover a wide range of microbial strains. The wide coverage of microbial strains is supported by recent research regarding antimicrobial efficiency of ZnO NPs, suggesting that they can be very efficient against Gram-negative pathogens, such as *Acinetobacter baumannii* [48], *E. coli*, Stenotrophomonas acidaminiphila, P. aeruginosa [49], but also Gram-positive species, such as S.aureus, Bacillus sp. [49], and microfungi (i.e., Candida albicans, Botrytis cinerea, and Penicillium expansum) [50].

## 4. Discussion

The intimate mechanism of action of ZnO NPs against bacteria cells is still unknown. Current research argues that this depends on the bacterial strain and also on the physico-chemical properties of the ZnO NPs. The proposed mechanisms of ZnO NPs antimicrobial activity are: (i) the change of membrane potential after NPs attachment, resulting in depolarization, and imbalance in the transport system and other membrane function; and (ii) NPs internalization and intracellular release of Zn(II) that can also lead to bacterial cell death, following binding to internal target molecules or reactive oxygen species release [10,51].

Besides their capacity to kill bacteria, a very useful aspect is in refererence to the toxicity to the human cells, as ZnO NPs were described as safe, non-toxic, and biocompatible at concentrations needed for killing bacteria [27]. ZnO is also used to strengthen polymeric bionanocomposites due to its high elastic strength. In that case, these NPs represent ideal candidates for nano-modified and mechanically resistant dressings used for skin repairing on the wound surface and promoting healing. Also, studies reported ZnO NPs as potential antitumor agents, showing selective toxicity against cancer cells [52]. Nontheless, researchers should be aware of the limits of using ZnO NPs with respect to the uncontrolled Zn(II) release, which could impose certain side effects, especially when used in high concentrations and systemic therapy [53].

We have developed a dressing containing ZnO NPs, intended for local application, also because recent studies show they can stimulate skin regeneration and wound healing by promoting anti-inflammatory mechanisms [54]. Many studies have evaluated the interaction between NPs and planktonic bacteria, but fewer have assessed NPs–biofilm interaction. To our knowledge this is one of the few studies to investigate the activity of ZnO NPs containing polymeric wound dressings against planktonic and biofilm-embedded clinical strains isolated from wound infection, some of them exhibiting important resistance phenotypes (MRSA, ESBL, VRE, MDR strains). Planktonic cells present a very different interaction with antibiotics, and posibly with nanoparticles, as compared to mature biofilms [55]. Previous studies focusing on traditional antibiotic challenges to bacteria found that the lower metabolic activity of biofilm cells can reduce the effectiveness of certain antibiotics [56]. Also, some studies that have evaluated the impact of NPs on biofilms have shown that biofilms, as compared to planktonic cells, show a reduced susceptibility to NPs [57].

The tested wound dressing was efficient both in Gram-positive and Gram-negative strains isolated from wound infections, including MDR isolates.

This study also revealed the fact that the bacterial viability is decreased in contact with ZnO nano-modified dressings and this phenotype is significant after 6 h of contact or more. Our study reports a higher efficiency of these dressings against Gram-negative strains, as compared to Gram-positive ones. This may be caused by differences in the cellular wall of these microorganisms, aspects previously reported for other NPs [58,59]. The low colonization potential of pathogenic bacteria of nano-ZnO surfaces is of great importance for the medical field, considering that surface colonization by viable bacteria is the first stage, absolutely necessary, to trigger the infectious process. Also, the loss of bacterial viability prevents the spreading of pathogenic strains and antibiotic-resistant bacteria in healthcare settings and communities.

*In vitro* experiments revealed that tested ZnO-containing dressing had a significant antimicrobial activity, their efficiency being influenced by the ZnO NPs concentration and the bacterial strain. Results highlight that the antibacterial activity was observed not only in planktonic cells but also in monospecific biofilm (a state in which bacterial cells are more tolerant to antibiotics or other antimicrobial substances). Moreover, the antibiofilm activity is maintained for up to 48 h or even 72 h in some difficult-to-eradicate opportunistic pathogens such as *P. aeruginosa.*

## 5. Conclusions

In this study we report the fabrication and characterization of a nano-coated wound dressing containing ZnO NPs to be evaluated with respect to antimicrobial potential in relevant wound pathogens. Mixing planktonic and biofilm bacteria models of clinically isolated resistant wound microorganisms represents the greatest innovation of our study, improving the understanding of the antimicrobial potential of ZnO NPs.

Obtained data show that the analyzed nano-coatings impair bacteria viability in a dose and time-dependent manner, being significantly efficient in bacteria-killing after 6 h of contact. Concentrations of 0.6% and 0.9% ZnO NPs were the most efficient loads for planktonic and biofilm growth inhibition in all tested laboratory and clinical isolates.

Such wound dressings could be efficiently applied to manage chronic wounds since their antimicrobial efficiency becomes significant after 6 h and is maintained for up to 3 days.

This report suggests that ZnO NPs containing dressings are efficient candidates for designing modern wound dressings tailored to prevent and heal biofilm infections and chronic wounds.

## Figures and Tables

**Figure 1 materials-14-03084-f001:**
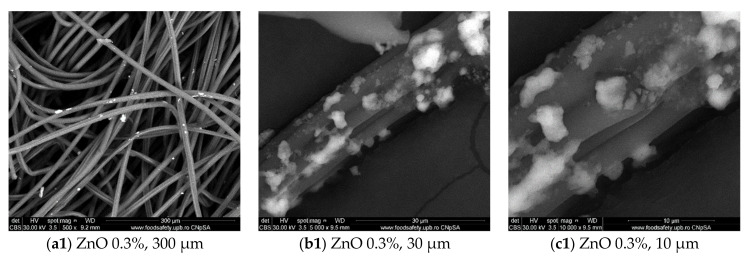
SEM images of the ZnO NPs-coated dressings (ZnO 0.3%; 0.6%; 0.9%) at various similar magnification ((**a1**–**a3**) = scale is 300 µm, (**b1**–**b3**) = scale is 30 µm, (**c1**–**b3**) = scale is 10 µm).

**Figure 2 materials-14-03084-f002:**
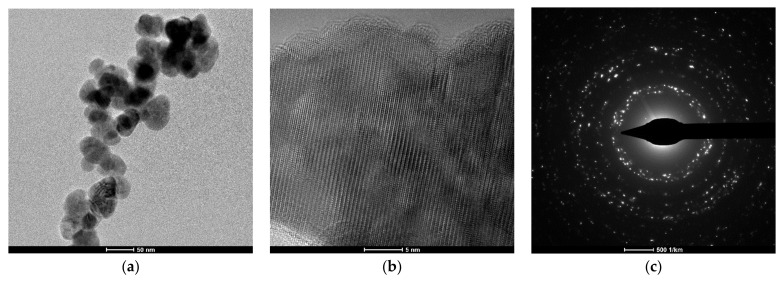
TEM images of the obtained ZnO NPs: (**a**) brigh-field; (**b**) HR-TEM and (**c**) SAED pattern.

**Figure 3 materials-14-03084-f003:**
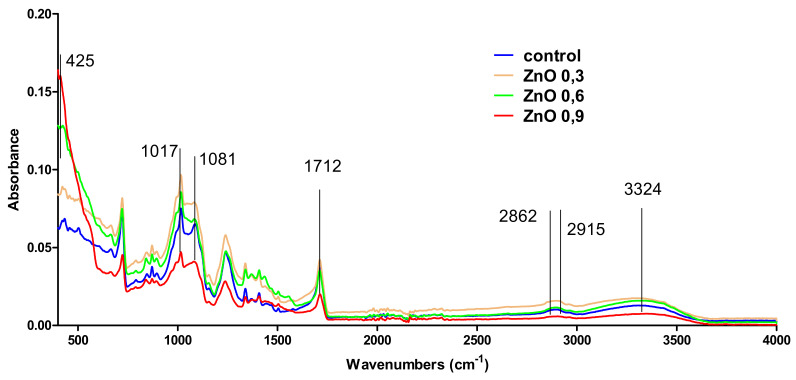
FT-IR spectra for control and for the ZnO NPs-coated dressings (ZnO 0.3; 0.6; 0.9%) and control.

**Figure 4 materials-14-03084-f004:**
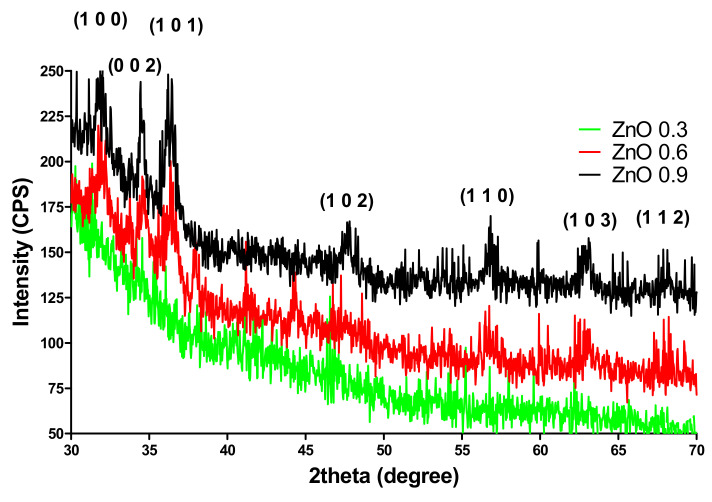
XRD spectra of ZnO NPs-coated dressings (ZnO 0.3; 0.6; 0.9%).

**Figure 5 materials-14-03084-f005:**
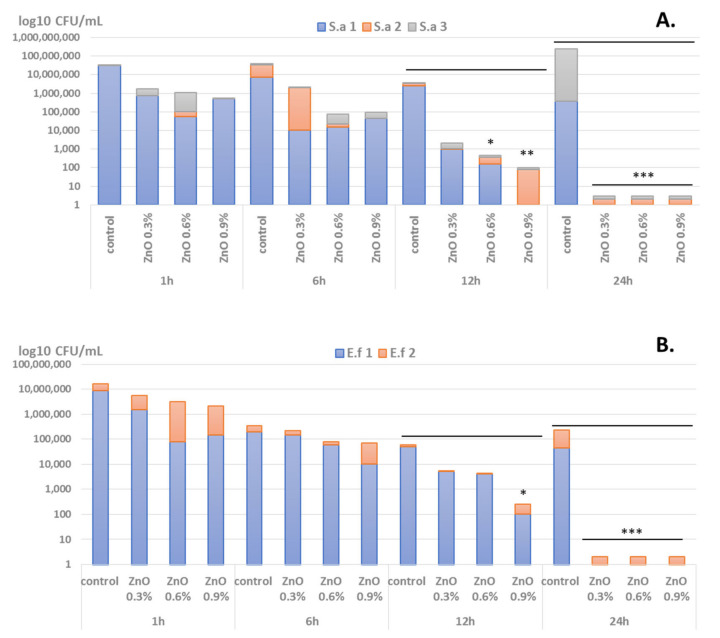
Graphical representation of log10 CFU/mL values obtained for the Gram-positive tested microbial strains (three *S. aureus (S.a.)* = (**A**) and *2 E. faecalis (E.f.)* isolates = (**B**)), expressing viability of bacteria incubated as 0.5 McFarland suspensions in PBS on the ZnO NPs-coated wound dressings for 1 h, 6 h, 12 h, and 24 h (* *p* < 0.05; ** *p* < 0.001; *** *p* < 0.0001 by comparing viability on control coatings with ZnO-coated dressings).

**Figure 6 materials-14-03084-f006:**
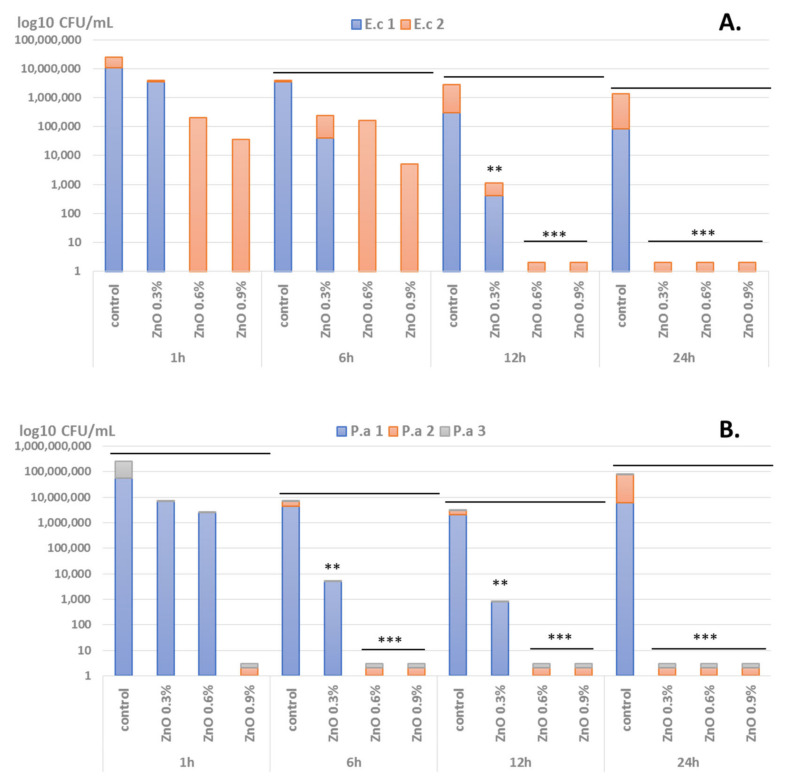
Graphical representation of log10 CFU/mL values obtained for the Gram-negative tested microbial strains (2 *E. coli (E.c.)* = (**A**) and 3 *P. aeruginosa* (P.a.) isolates = (**B**)), expressing viability of bacteria incubated as 0.5 McFarland suspensions in PBS on the ZnO NPs-coated wound dressings for 1 h, 6 h, 12 h, and 24 h (** *p* < 0.001; *** *p* < 0.0001 by comparing viability on control coatings with ZnO-coated dressings).

**Figure 7 materials-14-03084-f007:**
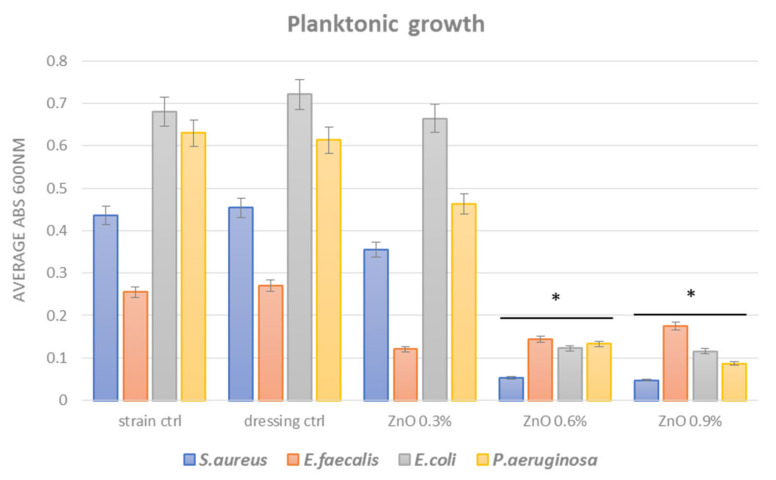
Graphic representation of average absorbances at 600 nm revealing growth of planktonic microbial cultures in the presence of control and nano-coated dressings for 24 h at 37 °C (* *p* < 0.05 by comparing average Abs 600 nm of all strains of the analyzed species growth in the presence of control and ZnO NPs containing coatings).

**Figure 8 materials-14-03084-f008:**
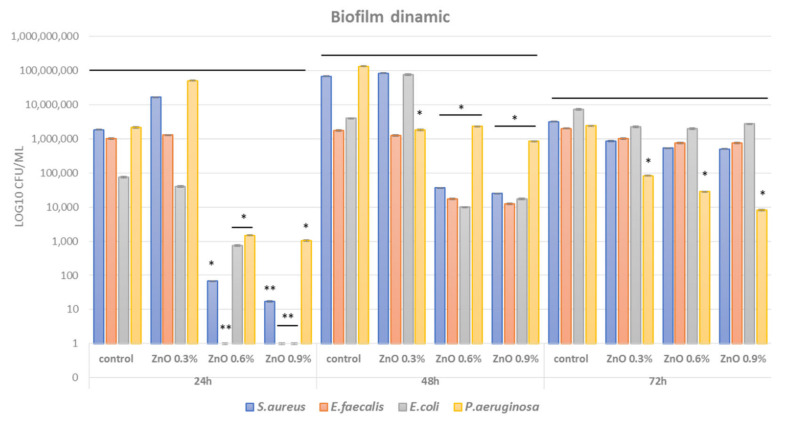
Graphic representation of average log10 CFU/mL values obtained for the tested Gram-positive and Gram-negative microbial strains, expressing biofilm embedded cells developed on control and ZnO NPs-coated dressings for different time points (24 h, 48 h, 72 h) (* *p* < 0.05; ** *p* < 0.001 by comparing biofilm formation on control and each ZnO-coated dressings).

**Table 1 materials-14-03084-t001:** Codes, source, and type of microbial strains utilized in this study.

Code	Microbial Strain	Type, Source
S.a 1	*S. aureus* ATCC 23235	Laboratory strain, American Type Cell Collection (ATCC)
S.a 2	*S.aureus 2*	Clinical wound infection isolate, MRSA (methicillin-resistant *S. aureus*)
S.a 3	*S.aureus 3*	Clinical wound infection isolate, MRSA
E.f 1	*Enterococcus feacalis* ATCC 29212	Laboratory strain, ATCC
E.f 2	*E. feacalis 2*	Clinical wound infection isolate, VRE (vancomycin-resistant enterococcus)
E.c 1	*E.coli* ATCC 25922	Laboratory strain, ATCC
E.c 2	*E. coli 2*	Clinical wound infection isolates, ESBL (extended-spectrum beta-lactamase)
P.a 1	*P. aeruginosa* ATCC 27853	Laboratory strain, ATCC
P.a 2	*P. aeruginosa 2*	Clinical wound infection isolate, MDR (multidrug-resistant strain)
P.a 3	*P. aeruginosa 3*	Clinical wound infection isolate, MDR (multidrug-resistant strain)

## Data Availability

All the data presented in this study is original and sufficient information regarding materials and methods was provided to ensure reproductibility of the research. Microbial strains and specimens of the nanomaterials described in the study are available on request from the corresponding author.

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
