# Peer review of "ZnO Nanoparticles-Modified Dressings to Inhibit Wound Pathogens"

_materials, 2021, doi:10.3390/ma14113084_

Round 1

Reviewer 1 Report

Recommendation: Major revision

Comments:

  • Line 17: What is meant by skin-related processes?
  • Line 60-61: what types of nanoparticles (metal, metal oxides, polymeric, etc) are the authors talking about? It should be mentioned with sufficient literature.
  • Page 64: Examples of metal/metal oxide NPs should be given. The authors can cover the following papers.
  • https://doi.org/10.1016/j.ceramint.2013.02.041
  • https://doi.org/10.1016/j.colsurfa.2011.12.011
  • https://doi.org/10.1016/j.jallcom.2016.02.067
  • The IR spectra should be redrawn by using Origin software.
  • TEM study is required to understand the morphology of ZnO.
  • Did the authors carry out any test to confirm that the prepared ZnO is not toxic?

Author Response

Dear reviewer,

Many thanks for your recommendation and comments. They greatly helped us to improve our paper.

Please find bellow the step-by -step reply to your comments.

  • Line 17: What is meant by skin-related processes?

Answer: we have reformulated

  • Line 60-61: what types of nanoparticles (metal, metal oxides, polymeric, etc) are the authors talking about? It should be mentioned with sufficient literature.

Answer: We have mentioned the nanoparticles and literature indications were given.

  • Page 64: Examples of metal/metal oxide NPs should be given. The authors can cover the following papers.
  • https://doi.org/10.1016/j.ceramint.2013.02.041
  • https://doi.org/10.1016/j.colsurfa.2011.12.011
  • https://doi.org/10.1016/j.jallcom.2016.02.067

Answer: Many thanks for your recommendation. We have read the above papers and found the information very useful for our research. Additional references were included.

  • The IR spectra should be redrawn by using Origin software.

Answer: done

  • TEM study is required to understand the morphology of ZnO.

Answer: done

  • Did the authors carry out any test to confirm that the prepared ZnO is not toxic?

Answer: we have started to investigate also the biocompatibility and biodistribution of the prepared NPs and coatings, in vitro and in vivo. These data will be presented in a future paper, after completion. Preliminary MTT assay show no toxicity if these ZnO NPs (data is not shown in this paper because here we focus on the antimicrobial potential of the nanoparticles).

Reviewer 2 Report

The manuscript materials-1238606, “ZnO nano-modified dressings to inhibit biofilm development in wound pathogens”, presents an interesting experimental study allowing to enrich the existing literature data on the antibacterial and antibiofilm-formation features of ZnO NPs, and their potential use for coating commercial wound dressings.

The strongest point of the manuscript is the range of microbial strains used in the present quantitative investigation of ZnO NPs effectiveness, i.e., including laboratory and clinical isolates of significant Gram-negative (Pseudomonas aeruginosa and Escherichia coli) and Gram-positive (Staphylococcus aureus and Enterococcus faecalis) opportunistic pathogens.

However, several parts of the manuscript need to be reformulated and reorganized, in order to highlight the originality and scientific accuracy of the present work. Please find some non-exhaustive suggestions in the attached file.

In conclusion, I recommend a major revision of the present manuscript.

Author Response

Dear reviewer,

Many thanks for your recommendation and comments. They greatly helped us to improve our paper.

Please find bellow the step-by -step reply to your comments.

  1. The introductory part generally presents the wound healing complications and the potential of ZnO
    nanoparticles as antibacterial and antibiofilm-formation.
    However, the sentence “[…] studies lack significant results showing the antimicrobial mechanisms,
    efficiency and impact in biofilm wound pathogens” (lines 84-85) is incorrect, as several mechanisms are
    already identified and presented in the literature - see at least the review “An overview on antimicrobial
    and wound healing properties of ZnO nanobiofilms, hydrogels, and bionanocomposites based on
    cellulose, chitosan, and alginate polymers”, Alavi M, Nokhodchi A, DOI 10.1016/j.carbpol.2019.115349.
    Furthermore, without presenting the role of the release of Zn2+ ions from ZnO NPs as a major
    mechanism for oligodynamic activities against eukaryotic and prokaryotic microorganisms, the
    manuscript is completely passing over the consequences of the uncontrollable release of Zn2+ ions in
    physiological conditions. Or, this migration is well-known and requires higher doses of ZnO NPs in wound
    dressings, with the direct risk to neutralize the therapeutic effects of formulation and to increase side
    effects of the ZnO-wound dressings. These aspects, and a critical comparison with the existing ZnO NPswound dressing systems and typical ZnO NPs concentrations, are essential, for pointing out the
    originality of the present study.
    In addition, the aim of the work is not completely clear, as the announced study is not generating new
    significant results showing the antimicrobial mechanisms of such ZnO NPs coated dressings
  • Answer: we have extensively revised our paper following your suggestions:
  • Answer: we have discussed and clarified the issues indicated by the reviewer (please see the last 2 paragraphs in Introduction) and also pointed out the novelty of the study.
  1. Please explain the interest of the dressing type (main characteristics) chosen to be coated with ZnO NPs.
    Which kind of interaction is expected between the polymer fiber and the NPs? Also, please explain the
    range of ZnO concentrations.

Answer: we have discussed and clarified the issues indicated by the reviewer (please see the last paragraph in Introduction).

  1. The Results part contains several paragraphs assessing literature data and not the present experimental
    results (see, for instance, lines 177-189). This kind of information normally belongs to Introduction.

Answer: we have reformulated and transferred this information to introduction or discussion sections.

  1. Paragraph 3.1.1. SEM starts with a conclusion-sentence, before even presenting the effective
    experimental results. Lines 192-194 should be reformulated and moved in a more appropriate place.

Answer: we have reformulated

  1. Figure 1 is not sufficiently clear and does not entirely support the text describing the SEM images.
    a. First, the scale is almost unreadable and not the same in ZnO 0.3 (a), ZnO 0.6 (a), ZnO 0.9 (a).
    b. Also, there is a question of representatively of the chosen images, as the dressing with ZnO 0.6
    appears to be better coated than the one with ZnO 0.9. So, is the coating with ZnO uniform
    enough?
    c. Please better explain the observations from 1.(b) and 1.(c) series – in which conditions the
    nanoparticles tend to agglomerates and why? For instance, the fiber with ZnO 0.3 (Fig. 1b)
    seems perfectly coated, better than the ones exposed to 0.6 and 0.9 ZnO NPs content.

Answer: We have selected SEM images of same scale and replaced old images, Also the scale was mentioned into the figure caption. We have explained the differences in agglomeration.

  1. Considering the Antimicrobial evaluation (paragraph 3.2.), the data from Figures 4, 5 and 6 are not fully
    explained and discussed. The dedicated paragraphs are mostly descriptive. A critical comparison with
    literature data is necessary.

Answer: we have improved the antimicrobial results section as suggested.
7. The Discussion part is once again a presentation of well-known facts from literature data instead of a
clear and pertinent discussion of the present experimental findings vs. those already reported in the
literature. Please also include the actual limits of using ZnO NPs – coated dressings, with respect to the
uncontrolled Zn2+ release and its side effects to the wounds.

Answer: we have improved the Discussion as suggested.

  1. The Conclusions should explicitly present the original results and their impact with respect to the
    already existing literature data on the same topic.

Answer: we have improved conclusion section.

Reviewer 3 Report

This study describes the fabrication and characterization of nano-coated dressing containing various amounts of ZnO NanoParticles (ZnO-NP) and the evaluation with respect to antimicrobial potential in four relevant wound pathogens including resistant nosocomial strains. Data indicate that ZnO-NPs impair bacteria viability in a dose and time-dependent manner for planktonic and biofilm cultures. Concentrations of 0.9% ZnO-NPs is the tested most efficient loads for growth inhibition, and 0.3% is sometimes effective depending on the strain. Altogether, it is suggested that ZnO-NPs containing dressings are efficient candidates for wound dressings tailored to treat biofilm infections and chronic wounds in order to minimize bacterial infections and to accelerate wound healing.

The manuscript is interesting, and it could become suitable for publication, but I think that it should be improved. Most of the points are related to the text and the style rather than to the experimental part. The following points should be addressed:  

First suggestion: about title modification “ZnO nano-modified dressings to inhibit biofilm development in wound pathogens. The word nanoparticles should be included. (2) biofilm is included in the title, but the study contains abundant data about planktonic cultures. Even more, at lines 269-270, it is written that Planktonic (free-floating) growth of microorganisms is important in the evaluation and management of wound infections. Thus, title is not totally appropriated.

Introduction

First, the complete description of the material, ZnO NPs coated polyester-nylon wound dressing, should be included at the beginning of the introduction.

In turn, authors should emphasize what is the new contribution of this manuscript. As authors mention (lines 70-72) zinc oxide (ZnO) nanoparticles have enhanced antimicrobial properties, being efficient against clinically relevant microbial pathogens, such as S. aureus [16], P.aeruginosa 71 [17] and Escherichia coli [18]. Furthermore (lines 82-83), it is written that recent studies show that ZnO NPs achieve successful wound closure and aesthetic wound healing in a skin wound mouse model [24].

What is the criteria for reference numbering? There is a jump from ref. 7 to ref. 38

Legend of Figure 1 Indicate quantitative magnification; low, intermediate and higher is not convincing. Indicate concentration units for 0.3, 0.6 and 0.9 (add %). This is also applied to Figure 2. Figure legends are por. At Figure 2, please indicate the meaning of the number in brackets inside the Figure.

According this figure, and according antimicrobial evaluation, ZnO-NPs 0.3% is not sufficient for detection and for antimicrobial effects. This should be clearly discussed or concluded at the end of the manuscript. Anyway, the critical point is: Is 0.9% the optimal concentration?. Is the antibacterial effect saturated? Can the wound dressing charged with higher amounts of ZnO-NPs?

Figure 4: Indicate that abbreviation of bacterial strains is according to Table 1. Indicate the meaning of A and B (on the right). Same for Figure 5. As a general idea, try to make Figure legends self-explicatory.

Line 285: Concerning 3.2.3. Biofilm modulation

This paragraph should be re-written. Although the description is partially correct, P. aeruginosa is the more resistant strain at 24h even in the presence of 0.9% ZnO-NPs. It seems a slow but long action on this bacterial strain, as a persistent effect. Persistent cells might be related to that pattern,  

Last paragraph at results, lines 302-308, it would be translated to the end of discussion or even a short conclusion. Sentences like “Further studies to reveal their in vivo efficiency and toxicity are necessary for the future to allow their use in clinical practice” are not appropriate in the results section.

Discussion, lines 309-313: I agree that the mechanism of action of ZnO is unknown. However, about proposed mechanisms, the first one (I) is not very likely. ROS are not associated to Zinc as Zn(II) is not a redox cation, such as Cu or Fe. ON the other hand, Zn(II) is related to transcription factors containing Zn fingers, metalloproteases or stimulation of immune system. In summary, proposal (i) should be deleted, or alternative references are needed.

Unification of point (not comma) at decimal numbers is needed. It is detected sometimes 0.3 and sometimes 0,3 % (i.e lines 206, 245).

Zn is Zn(II), repair Zn+ at line 316

Idea at line 324

Author Response

Dear reviewer,

Many thanks for your recommendation and comments. They greatly helped us to improve our paper.

Please find bellow the step-by -step reply to your comments.

This study describes the fabrication and characterization of nano-coated dressing containing various amounts of ZnO NanoParticles (ZnO-NP) and the evaluation with respect to antimicrobial potential in four relevant wound pathogens including resistant nosocomial strains. Data indicate that ZnO-NPs impair bacteria viability in a dose and time-dependent manner for planktonic and biofilm cultures. Concentrations of 0.9% ZnO-NPs is the tested most efficient loads for growth inhibition, and 0.3% is sometimes effective depending on the strain. Altogether, it is suggested that ZnO-NPs containing dressings are efficient candidates for wound dressings tailored to treat biofilm infections and chronic wounds in order to minimize bacterial infections and to accelerate wound healing.

The manuscript is interesting, and it could become suitable for publication, but I think that it should be improved. Most of the points are related to the text and the style rather than to the experimental part. The following points should be addressed:  

First suggestion: about title modification “ZnO nano-modified dressings to inhibit biofilm development in wound pathogens. The word nanoparticles should be included. (2) biofilm is included in the title, but the study contains abundant data about planktonic cultures. Even more, at lines 269-270, it is written that Planktonic (free-floating) growth of microorganisms is important in the evaluation and management of wound infections. Thus, title is not totally appropriated.

 Answer: we have revised the title as suggested

Introduction

First, the complete description of the material, ZnO NPs coated polyester-nylon wound dressing, should be included at the beginning of the introduction.

In turn, authors should emphasize what is the new contribution of this manuscript. As authors mention (lines 70-72) zinc oxide (ZnO) nanoparticles have enhanced antimicrobial properties, being efficient against clinically relevant microbial pathogens, such as S. aureus [16], P.aeruginosa 71 [17] and Escherichia coli [18]. Furthermore (lines 82-83), it is written that recent studies show that ZnO NPs achieve successful wound closure and aesthetic wound healing in a skin wound mouse model [24].

Answer: the novelty of the study was pointed out in the paper. (We have developed a dressing containing ZnO NPs, since rescent studies show they can stimulate skin regeneration and wound healing by an anti-inflammatory mechanisms [28]. Many studies have evaluated the interaction between NPs and planktonic bacteria, but few have assessed NPs–biofilm interaction. To our knowledge this is the first study to investigate the activity of ZnO NPs containing polymeric wound dressings against planktonic and biofilm embedded clinical strains of microorganisms isolated from wound infection. planktonic cells present a very different interaction with antibiotics, and posibly with nanoparticles, as compared to mature biofilms [29]. Previous studies focusing on traditional antibiotic challenges to bacteria found that the lower metabolic activity of biofilm cells can reduce the effectiveness of certain antibiotics [30]. Also, some studies that have evaluated the impact of NPs on biofilms, have shown that biofilms, as compared to planktonic cells, shown a reduced susceptibility to NPs [31].)

What is the criteria for reference numbering? There is a jump from ref. 7 to ref. 38

 Answer: we revised references and numbered accordingly

Legend of Figure 1 Indicate quantitative magnification; low, intermediate and higher is not convincing. Indicate concentration units for 0.3, 0.6 and 0.9 (add %). This is also applied to Figure 2. Figure legends are por. At Figure 2, please indicate the meaning of the number in brackets inside the Figure.

According this figure, and according antimicrobial evaluation, ZnO-NPs 0.3% is not sufficient for detection and for antimicrobial effects. This should be clearly discussed or concluded at the end of the manuscript. Anyway, the critical point is: Is 0.9% the optimal concentration?. Is the antibacterial effect saturated? Can the wound dressing charged with higher amounts of ZnO-NPs?

Answer: We have selected SEM images of same scale and replaced old images, Also the scale was mentioned into the figure caption. We have explained the differences in NPs agglomeration, depending on their concentration.

Figure 4: Indicate that abbreviation of bacterial strains is according to Table 1. Indicate the meaning of A and B (on the right). Same for Figure 5. As a general idea, try to make Figure legends self-explicatory.

 Answer: we have indicated the meaning of A and B in the mentioned figures.

Line 285: Concerning 3.2.3. Biofilm modulation

This paragraph should be re-written. Although the description is partially correct, P. aeruginosa is the more resistant strain at 24h even in the presence of 0.9% ZnO-NPs. It seems a slow but long action on this bacterial strain, as a persistent effect. Persistent cells might be related to that pattern,  

 Answer: we have improved the results and discussions as requested

Last paragraph at results, lines 302-308, it would be translated to the end of discussion or even a short conclusion. Sentences like “Further studies to reveal their in vivo efficiency and toxicity are necessary for the future to allow their use in clinical practice” are not appropriate in the results section.

 Answer: we have reformulated or deleted inappropriate sentences

Discussion, lines 309-313: I agree that the mechanism of action of ZnO is unknown. However, about proposed mechanisms, the first one (I) is not very likely. ROS are not associated to Zinc as Zn(II) is not a redox cation, such as Cu or Fe. ON the other hand, Zn(II) is related to transcription factors containing Zn fingers, metalloproteases or stimulation of immune system. In summary, proposal (i) should be deleted, or alternative references are needed.

 Answer: we have reformulated

Unification of point (not comma) at decimal numbers is needed. It is detected sometimes 0.3 and sometimes 0,3 % (i.e lines 206, 245).

Answer: we have corrected

Zn is Zn(II), repair Zn+ at line 316

Answer: we have corrected

Idea at line 324

Answer: we have corrected

Round 2

Reviewer 1 Report

The paper can be accepted.

Reviewer 2 Report

The revised manuscript is largely improved.

I recommend it for publication in the present form.

Reviewer 3 Report

Authors address all points related to my previous report. In fact, the manuscript has been modified significantly to address reports from other  reviewers. As far as I concern, I think that the article has been improved in presentation and content. Some points at the discussion could not totally convincing yet, but this is time to stop further review. Methods are correct and appropriate, the paper is well referenced.